# Automated Large-Scale Production of Paclitaxel Loaded Mesenchymal Stromal Cells for Cell Therapy Applications

**DOI:** 10.3390/pharmaceutics12050411

**Published:** 2020-04-30

**Authors:** Daniela Lisini, Sara Nava, Simona Frigerio, Simona Pogliani, Guido Maronati, Angela Marcianti, Valentina Coccè, Gianpietro Bondiolotti, Loredana Cavicchini, Francesca Paino, Francesco Petrella, Giulio Alessandri, Eugenio A. Parati, Augusto Pessina

**Affiliations:** 1Cell Therapy Production Unit-UPTC and Cerebrovascular Diseases Unit, Fondazione IRCCS Istituto Neurologico Carlo Besta, 20133 Milan, Italy; sara.nava@istituto-besta.it (S.N.); simona.frigerio@istituto-besta.it (S.F.); simona.pogliani@istituto-besta.it (S.P.); a.marcianti@campus.unimib.it (A.M.); giulio.alessandri@istituto-besta.it (G.A.); eugenio.parati@istituto-besta.it (E.A.P.); 2Synlab CAM Polidiagnostico, 20900 Monza, Italy; guido@maronati.it; 3CRC StaMeTec, Department of Biomedical, Surgical and Dental Sciences, University of Milan, 20100 Milan, Italy; valentina.cocce@guest.unimi.it (V.C.); Loredana.cavicchini@unimi.it (L.C.); francesca.paino@unimi.it (F.P.); augusto.pessina@unimi.it (A.P.); 4Department of Medical Biotechnology and Translational Medicine, University of Milan, 20100 Milan, Italy; gianpietro.bondiolotti@unimi.it; 5Division of Thoracic Surgery, IEO, European Institute of Oncology IRCCS, 20141 Milan, Italy; francesco.petrella@ieo.it; 6Department of Oncology and Hemato-Oncology, University of Milan, 20100 Milan, Italy

**Keywords:** cell expansion, cell therapy, GMPs, MSCs, paclitaxel

## Abstract

Mesenchymal stromal cells (MSCs) prepared as advanced therapies medicinal products (ATMPs) have been widely used for the treatment of different diseases. The latest developments concern the possibility to use MSCs as carrier of molecules, including chemotherapeutic drugs. Taking advantage of their intrinsic homing feature, MSCs may improve drugs localization in the disease area. However, for cell therapy applications, a significant number of MSCs loaded with the drug is required. We here investigate the possibility to produce a large amount of Good Manufacturing Practice (GMP)-compliant MSCs loaded with the chemotherapeutic drug Paclitaxel (MSCs-PTX), using a closed bioreactor system. Cells were obtained starting from 13 adipose tissue lipoaspirates. All samples were characterized in terms of number/viability, morphology, growth kinetics, and immunophenotype. The ability of MSCs to internalize PTX as well as the antiproliferative activity of the MSCs-PTX in vitro was also assessed. The results demonstrate that our approach allows a large scale expansion of cells within a week; the MSCs-PTX, despite a different morphology from MSCs, displayed the typical features of MSCs in terms of viability, adhesion capacity, and phenotype. In addition, MSCs showed the ability to internalize PTX and finally to kill cancer cells, inhibiting the proliferation of tumor lines in vitro. In summary our results demonstrate for the first time that it is possible to obtain, in a short time, large amounts of MSCs loaded with PTX to be used in clinical trials for the treatment of patients with oncological diseases.

## 1. Introduction

Mesenchymal stromal cells (MSCs) are considered a promising tool for cell therapy in both regenerative medicine and immune diseases such as Graft-versus-Host Disease (GvHD), Crohn’s Disease (CD), and Rheumatoid Arthritis (RA) [1]. Until 2016, 493 MSCs-based clinical trials, either complete or ongoing, appeared in the database of the US National Institutes of Health [2]. Recently MSCs have been indicated as a potential new important tool for delivering chemotherapeutic drugs. In fact, it has been reported that MSCs, upon in vitro exposure to very high concentrations of Paclitaxel (PTX) can uptake significant amounts of the drug, subsequently released at concentrations high enough to strongly inhibit cancer cell proliferation, when the PTX loaded MSCs (MSCs-PTX) were located nearby. We previously demonstrated that the uptake of PTX by MSCs occurs without significant signs of cell toxicity; PTX affects cytoskeleton by promoting microtubule polymerization that induced the mitotic arrest of the cells. Morphology and sub-cellular organization of MSCs-PTX were similar to that of MSCs, but an increased number of vacuole-like structures was detected in MSCs-PTX, some of which were attributable to microvescicles (MV). PTX, upon entering the cells, locates mainly into MV derived from Golgi apparatus, where it remains until the moment of the PTX release, that would occur by fusion of MV membrane with cell membrane, permitting the secretion of PTX in the extracellular environment and the subsequent antitumor activity. Moreover, the potent anticancer activity of MSCs-PTX has been proven both in vitro and in vivo using different experimental cancer models [3,4,5,6,7,8,9,10]. Clavreul and coauthors also demonstrated that MSCs are able to uptake Sorafenib (SFN), carry this drug to a brain tumor via intranasal administration, and release SFN, resulting in a lower levels of tumor angiogenesis. The mechanisms by which MSCs uptake/release SFN did not seem to cause cell toxicity, because 80% of MSCs remained viable 7 days after priming with drug. The route by which SFN leaves MSCs is not completely known, but as for PTX, it seems that SFN locates into MV and drug release seems to be connected to the secretion of MV through cell membrane [11].

The use of MSCs in cell therapy approaches requires a rapid large-scale expansion of cells under Good Manufacturing Practice (GMP) rules, to obtain a clinically relevant number. Traditionally, MSCs preparation for clinical use can present some critical aspects: (a) cell expansion requires the use of a large number of flasks; (b) long time in incubators installed in cleanroom facilities; (c) extensive manipulation by the operators with high risk of microbial contamination, and (d) high cost for GMP-compliant cells preparation.

With cell-based therapies moving towards commercialization and the increase of clinical trials in late stage development, it is clear that the selection of suitable manufacturing processes is getting increasingly important [12]. In the last years several expansion technologies to achieve clinical-scale MSCs manufacture were investigated, including stacked and multi-layered flask systems, as well as fully automated bioreactors. The use of bioreactors presents many advantages if compared to cell expansion methods using culture flasks. A bioreactor, being a closed system, increases the safety of the process, reduces the risk of microbial contamination and the time of MSCs expansion, and finally allows to obtain a very high number of cells (due to the maximization of the surface area for MSCs growth) in a very short time. Moreover, the use of closed systems allows advanced therapy medicinal products (ATMPs) manufacturing in environments with less stringent GMP classification, thereby reducing production costs. In addition, bioreactors seem to guarantee a good quality of the product, because the procedure does not alter the phenotypic profile and functional activity of the cells. To note, the change of the culture methods can introduce modification of the characteristics and functionality of the cell products, as the properties of the cells may change even upon minor manipulations. Therefore, it is crucial to verify whether the method used for cell preparation results in a product that is comparable in terms of identity, safety, and potency [13,14,15,16,17].

On this basis, we here investigated the possibility to efficiently generate a large amount of MSCs loaded with the chemotherapeutic drug PTX, by using a novel, closed bioreactor system (Quantum Cell Expansion System, Terumo BCT), designed for automated cell expansion. Results demonstrated, for the first time, that our technical approach is efficient in terms of quantity and quality of MSCs-PTX obtained. We propose our innovative procedure to prepare MSCs-PTX as ATMP for cancer treatment in humans.

## 2. Materials and Methods

### 2.1. Sample Collection

Adipose tissue (AT) lipoaspirates were collected, under general anesthesia, from *n* = 13 healthy volunteer donors undergoing plastic surgery for aesthetic purposes. The mean age was 42.1 (range: 18–66). Samples were collected after signed informed consent of no objection for the use for research of surgical tissues (otherwise eliminated) in accordance with the Declaration of Helsinki. The informed consents were obtained prior to tissue collection; the Ethics Commettee of Regione Lombardia, Institutional Review Board Section of the IRCCS Neurological Institute C. Besta Foundation approved (Verbal Number 29, 4 May 2016) the design of the study.

Samples were processed within 24 h from surgery.

### 2.2. MSCs Isolation from Human Adipose Tissue

MSCs from AT lipoaspirates (AT-MSCs) were isolated as follows: the sample was disaggregated by enzymatic digestion with 200 U/mL of collagenase type I (Life Technologies, Carlsbad, CA, USA), then was centrifuged (1000× *g*, 15 min) and the floating fraction and cellular pellet were plated on 150 cm^2^ flasks (Euroclone, Milano, Italy), 10 mL/flask, and expanded in DMEM low glucose (Euroclone, Milano, Italy) supplemented with 5% platelet lysate Stemulate (Cook Reagent, Indianapolis, IN, USA) and 2 mM l-glutamine (Euroclone, Milano, Italy) until a number of at least 20 × 10^6^ at a passage not exceeding P3. Primary cultures were analyzed for number and viability at each passage, their population doubling time (PDT), and the expression of the typical MSC markers (CD90, CD73, CD105; all the monoclonal antibodies were provided from Becton Dickinson, Franklin Lake, NJ, USA), as described previously [18].

### 2.3. MSCs Expansion in Quantum Cell Expansion System

The Quantum Cell Expansion System (Terumo BCT, Lakewood, CO, USA) consists of a synthetic hollow fiber bioreactor (surface area of 2.1 m^2^) that works in a sterile closed-loop circuit for media and gas exchange, as described in details by other previous reports [19,20,21]. Briefly, after priming of the disposable expansion set (Terumo BCT, Lakewood, CO, USA), the bioreactor was coated overnight with 5 mg of human fibronectin (Corning Incorporated, Deeside, UK) to promote cell adhesion; a 4 L media bag was then attached to the appropriate inlet line on the Quantum disposable expansion set. All disposable bags used were provided by Terumo BCT, Lakewood, CO, USA.

At least 20 × 10^6^ MSCs expanded as described above were seeded in the system at an inlet rate of 25 mL/min, followed by a 24 h phase in which MSCs recirculate and adhere to the support. As cells are not visible in the hollow fiber, cell growth was estimated according to lactate generation by the cells in the system. Fresh complete media is added continuously to cells and the inlet rate is adjusted as required by the rate lactate generation. During the early phase of MSCs expansion the inlet rate was 0.1 mL/min up to a lactate value of 3 mM, then the inlet rate was increased at 0.2 mL/min up to a lactate value of 4 mM, 0.4 mL/min up to a lactate value of 5 mM, 0.8 mL/min up to a lactate value of 6 mM, and finally 1.2 mL/min. At the end of the expansion phase the cells were harvested or cultured for another 24 h for the drug loading phase.

### 2.4. Drug Loading Phase of MSCs with PTX

The drug loading of MSCs was performed in the bioreactor by priming the cells according to a procedure standardized in flasks as previously described, with some modifications [3,22,23]. Drug loading phase begin when lactate value increased less than 0.5 mM in 24 h.

A 4-L complete medium bag, supplemented with PTX at a final concentration of 10 µg/mL (PTX, TEVA, Milano, Italy, 6 mg/mL) was prepared and connected to the instrument, as described above. Through the “touch screen” the medium present in the instrument was completely replaced with medium supplemented with PTX, within 5 min. This phase was followed by the recirculation of the medium in the Quantum disposable set for 24 h at an inlet rate of 0.1 mL/min. After 24 h MSCs loaded with PTX were washed with PBS (Euroclone, Milano, Italy) and detached from the expansion set using recombinant trypsin (TrypLE Select 1X, Thermofisher, Waltham, MA, USA) and eluted in a 500 mL bag using complete medium.

An aliquot of fresh MSCs-PTX was used to determine cell number, viability by trypan blue exclusion test, as well as the quantity of PTX loaded by the cells. The MSCs-PTX were frozen in a sterile solution of NaCl 0.9% (pharmaceutical grade, Fresenius KABI, Lake Zurich, IL, USA) supplemented with 5% human albumin (pharmaceutical grade) and 10% of Dimethyl Sulfoxide (Cryosure DMSO GMP grade, Li StarFISH, Milano, Italy), and cryopreserved in 2 mL cryovials at the concentration of 40–45 × 10^6^ live cells per vial, 2 mL/vial, in nitrogen vapors.

A vial of 10 × 10^6^ cells was used to characterize MSCs-PTX after thawing.

Viability of MSCs-PTX, both fresh and after freeze/thawing, was assessed also over time, re-seeding cells in T25 flasks (three flasks/condition, total six flasks/experiment, *n* = 3 experiments) and the MSCs-PTX viability was analyzed after 7, 14, and 21 days. In this period of time medium was changed every 3 days; the cells had never been detached, due to the loss of their duplication capacity and the failure to reach confluence.

### 2.5. Annexin V and PI Staining

MSCs and MSCs-PTX were collected by centrifugation and washed twice with cold PBS. After careful remove of supernatant, cells were re-suspended in 1× Binding buffer, following manufacturer’s instruction, at a concentration of 1 × 10^6^ cells/mL, at least 100 µL per sample. Annexin V antibody and PI (Becton Dickinson, Franklin Lake, NJ, USA) were added to the samples and incubated for 20 min at room temperature in the dark. After incubation 400 μL of Binding buffer was added to each tube. Samples were analyzed immediately (within 1 h) by flow cytometry, using the instrument FACScalibur and the CellQuest Software (Becton Dickinson, Franklin Lake, NJ, USA). The data were interpreted as follow: Annexin V negative-PI negative populations are healthy cells; Annexin V positive-PI negative populations represent cells in early apoptosis; Annexin V positive-PI positive staining indicates necrotic cells (post-apoptotic necrosis or late apoptosis).

### 2.6. Tumor Cell Line

Human pancreatic adenocarcinoma cell line CFPAC-1 [24,25] was provided by Centro Substrati Cellulari, ISZLER (Brescia, Italy), the mesothelioma cell line (NCI H2052) [26] was kindly provided by Prof Roberta Alfieri (Clinical and Experimental Medicine Department, University of Parma, Italy). CFPAC-1 cells were maintained in complete medium (Iscove modified Dulbecco’s medium IMDM) supplemented with 10% Fetal Bovine Serum (FBS) by 1:5 weekly dilution, as mesothelioma cell line was cultured in RPMI 1640 Medium supplemented with 10% FBS, 1% Hepes, and 1% sodium pyruvate. All reagents were provided by Euroclone, Milano, Italy.

### 2.7. HPLC Analysis

The presence of PTX in the MSCs was demonstrated by a validated bioanalytical reversed phase high performance liquid chromatography (HPLC) assay, as previously described [27]. MSCs-PTX lysates (MSCs-PTX/LYS) were obtained by sonication performed by three cycles of 0.4 s pulse at 30% amplitude each (Labsonic UBraun, Reichertshausen, Germany), followed by centrifugation at 2500× *g* for 10 min.

For HPLC analysis MSC lysates (MSCs/LYS) were mixed (1/4 *v*/*v*) with ethyl acetate, vortexed for 8 min, and centrifuged. The supernatants were dried under vacuum (Rotavapor R 110, Büchi) and the residue reconstituted with 7 µL of mobile phase, filtered through 0.2 μm nylon filters (Phenomenex, Phenex-NY 4 mm) and an aliquot of 40 μL injected in HPLC. The chromatographic system (Agilent 1100 Series, Agilent Technologies, Inc. Santa Clara, CA, USA) equipped with nucleodur C_18_ column, 4.6 × 150 mm, 5 μm particle sizes (Macherey-Nagel), was operated at 30 °C in isocratic mode using acetonitrile and 0.1 M Ammonium Acetate (50/50 *v*/*v*) as mobile phase that was pumped at rate of 0.7 mL/min, the eluent was monitoring using UV–Visible DAD detector at 238 nm. The retention time of PTX was 11 min. A calibration curve (10–25–50–100 ng) of PTX in drug-free lysate was prepared and processed with the samples and used to quantify PTX (*y* = 1.1809*x* − 2.9565; R^2^ = 0.9963). The extraction recovery of PTX measured in calibration curve was 78%.

### 2.8. In Vitro Anticancer Activity of MSCs-PTX

To measure the amount of internalized drug, MSCs-PTX were washed twice with Hank’s solution (HBSS, Euroclone, Milano, Italy) and 3 × 10^6^ cells suspended in 1.5 mL of complete medium. The cells were lysed as previously described and MSCs-PTX/LYS were tested for their antiproliferative activity on standard cancer cell line CFPAC-1 in 96 multiwell plates (Sarstedt, Numbrecht, Germany) as previously described [27,28]. The activity of lysates was compared with that of pure PTX, according to a biological dosage assay and the lysate obtained from untreated MSCs were used as control. Briefly, 1:2 serial dilutions of pure drug (PTX), MSCs/LYS, and MSCs-PTX/LYS were performed in 100 µL of culture medium/well and then 1000 tumor cells were added to each well. The cell growth was evaluated after 7 days of culture at 37 °C and 5% CO_2_, by measuring the optical density at 550 nm in a MTT (3-(4,5-dimethyl-2-thiazolyl)-2,5-diphenyl-2-H-tetrazoliumbromide) assay (Sigma-Aldrich, Darmstadt, Germany) [29]. The inhibitory concentrations (IC_50_) were determined as µL/well of lysate according to the Reed and Muench formula [30]. By comparing the IC_50_ value of lysate with that of pure PTX (ng/mL) the PTX Equivalent Concentration (PEC) was extrapolated: PEC (ng/mL) = IC_50_ PTX × 100/V_50_ (µL/well) where IC_50_ PTX = the concentration of pure PTX and V_50_ = the volume of MSCs-PTX/LYS able to produce a 50% inhibition of cell proliferation. The PEC value was used to calculate the amount of PTX incorporated by a single MSC (pg/cell) = PEC (ng/mL) × lysate volume (mL) × 1000/number of cells [31].

### 2.9. Potency Test

Conditioned media from MSCs-PTX (MSCs-PTX/CM) were prepared seeding MSCs-PTX in 48 multiwell plates (24,000 cells/well, 350 µL/well); the plates were maintained at 37 °C, 5% CO_2_, and Conditioned Media (CM) were collected after 24, 48, 72, 96, 120 h after seeding. The effect of pure PTX and MSCs-PTX/CM against tumor cell proliferation was studied in 96 multiwell plates (Sarstedt, Numbrecht, Germany) by using as target pancreatic adenocarcinoma cells (CFPAC-1) and a mesothelioma cell line (NCI H2052). Briefly, 10^3^ tumor cells were seeded in each well in 100 µL of culture medium/well and after 24 h MSCs-PTX/CM was added to each well. A serial dilutions curve of pure drug as well as MSCs-PTX/LYS were plated as controls. After 7 days of culture at 37 °C and 5% CO_2_ cell growth was evaluated by MTT assay (Promega, Medison, WI, USA), as previously described [29].

### 2.10. Statistical Analysis

Data are expressed as the mean ± SD. For statistical analysis *t*-test and one-way analysis of variance (ANOVA) using GraphPad INSTAT program (GraphPad Software Inc., San Diego, CA, USA) and *p* values < 0.05 were considered statistically significant. The linearity of response and the correlation were studied using regression analysis by Excel Software, Version 2007 (Microsoft, Inc., Redmond, WA, USA).

## 3. Results

### 3.1. MSCs Isolation from Human Adipose Tissue

The mean quantity of the 13 lipoaspirate samples was 40.55 ± 11.3 mL (mean ± SD, range from 25 to 60 mL).

We were able to isolate MSCs from 100% of the samples. MSCs sprouted from the lipoaspirates after a median time of 3.88 days (range: 3–5 days) and underwent the first detachment (Passage 1, P1) after a median time of 11.44 days (range: 7–14 days). The cells were expanded in flasks until a number of at least 20 × 10^6^, as requested in user’s instruction of Quantum Cell Expansion System, at a maximum passage of P3. The mean number of MSCs obtained after the first expansion was 28.5 × 10^6^ (SD: 3.96 × 10^6^).

The cultured cells displayed the typical spindle-shaped morphology (Figure 1A), with a PDT at P2 of 34.84 ± 12.07, at P3 of 33.54 ± 4.96 h.

During this culture period, cells maintained a high percentage of viability, as evaluated by trypan blue exclusion test at every passage; the mean value was 96.04%, (ranging from 93.18% to 99.42%). Importantly, cell viability was not affected by cryopreservation, as after thawing cells viability was 94.75% (mean), ranging from 92.15% to 98.56%.

The flow cytometry analysis of the cells confirmed the MSCs phenotype (see Section 3.3). 

### 3.2. MSC Expansion and Loading with PTX in Quantum System

A mean of 23.51 × 10^6^ MSCs (SD: 4.43 × 10^6^) were loaded for expansion in the Quantum Cells Expansion System. The first run of the bioreactor was performed in order to set a standard procedure for MSCs expansion, without loading cells with PTX; the expansion length from cell loading to the harvest phase was 6 days. During this period of time lactate value increased from 0.4 to 9.2 mM reached at the end of the expansion phase. The total number of recovered MSCs was 480 × 10^6^, with a viability of 94%.

After this preliminary experiment, 12 MSCs preparations were expanded and directly loaded with PTX in the bioreactor. After a mean of 6 days from start of expansions (range: 5–8 days), when lactate mean value was 8.07 mM (range: 5.5–10.3 mM) PTX was added and 24 h later cells were harvested. A mean of 591.43 × 10^6^ MSCs-PTX (range: 290 × 10^6^–940 × 10^6^ cells) with a mean viability of 94.34% were obtained. MSCs-PTX were then processed for cryopreservation.

MSCs-PTX viability was not affected by cryopreservation, as the mean viability after thawing was 91.4% (range 88.84%–95.5%); recovery after thawing was 83.35% ± 11.73% (mean ± SD).

Viability values of MSCs-PTX, both fresh and after freeze/thawing, were maintained over time up to 21 days after re-seeding of cells, as shown in the Appendix A.

### 3.3. Phenotypic Profile of MSCs-PTX

The MSCs-PTX recovered after thawing were investigated to verify if PTX treatment could have altered their phenotypic and functional features. Results indicate that MSCs-PTX maintained unchanged adhesion capacity to plastic support and showed an altered morphology, as expected (Figure 1B); moreover, cells lost the growth capacity.

The change in cell morphology does not seem to be linked to the bioreactor expansion process, since the same altered morphology also appears after loading MSCs with PTX in flasks (Appendix A).

Only a small percentage of MSCs-PTX displayed necrotic and/or apoptotic features but not significant differences compared to control untreated MSCs were found: necrotic cells were 9.9% ± 3.6% vs. 4.83% ± 0.2% (mean ± SD), apoptotic cells were 13.2% ± 11.72% vs. 5.2% ± 5.64% (mean ± SD) (Figure 2).

Results of the apoptotic and necrotic cell populations detected by Annexin V-FITC and PI staining of a representative sample of MSCs and the corresponding MSCs-PTX are shown in the Appendix A.

Flow cytometry analysis showed that MSCs, after the first expansion in flasks displayed high percentages of the typical MSC markers (CD90 = 94.4% ± 8.2%, CD105 = 80.5% ± 15.4%, and CD73 = 98.5% ± 1.2%) and were negative for hematopoietic markers (CD31 = 2.16% ± 2%, CD34 = 1.4% ± 1.7%, and CD45 = 11.5% ± 9.3%; these percentages were maintained after large-scale expansion and loading with PTX in bioreactor (CD90 = 88.3% ± 6%, CD105 = 83.7% ± 14.3%, CD73 = 98.9% ± 0.7%, CD31 = 1.2% ± 1.6%, CD34 = 3.68% ± 2.3%, and CD45 = 9.4% ± 5.1%) (Figure 3).

In the Appendix A the flow cytometry analysis of a representative MSCs line is displayed, before (A) and after (B) expansion and loading with PTX in the bioreactor.

### 3.4. HPLC Dosage of PTX Incorporated by MSCs

The drug incorporation of PTX by loaded MSCs was confirmed by HPLC analysis performed on nine samples. As shown by a standard HPLC chromatograms (1.000 ng/mL of PTX) (Figure 4A) the drug was eluted with a peak at 11 min. A peak with an identical retention time for PTX was eluted by processing MSCs-PTX/LYS (Figure 4B). HPLC analysis revealed the presence of other non-specific peaks (at 2.5–4 min) due to compounds produced by cells that do not correlate with the presence of PTX, these peaks also being present in the chromatogram of MSCs/LYS, used as control (Figure 4C). Figure 4D represents the standard chromatogram of PTX. The presence of the main PTX metabolite (6 alpha-hydroxypaclitaxel), normally eluted at 5.5 min, and of other PTX metabolites can be excluded (*p* > 0.5) [32].

### 3.5. In Vitro Anticancer Activity of MSCs-PTX

The anticancer activity of MSCs-PTX was evaluated by testing their cell lysate on the tumor cell line CFPAC-1, used as reference cells to validate the antitumor efficacy [33]. The antiproliferative effect of different dilutions of the cell lysates from cryopreserved MSCs-PTX (MSCs-PTX/LYS-FT), fresh MSCs-PTX (MSCs-PTX/LYS-F), and untreated MSCs obtained through bioreactor procedure were compared (Figure 5). Both MSCs-PTX/LYS-F and MSCs-PTX/LYS-FT induced a significant inhibition of CFPAC-1 proliferation showing a dose–response kinetics that was expressed as percentage of proliferation normalized on the basal value produced by control lysates, obtained from untreated MSCs. A significant linear regression slope (*p* < 0.001) and high coefficient of correlation (R^2^) were found (Figure 5A). As reported in Figure 5B,C, the biological dosage performed by comparing the dose–response inhibition curve of MSCs-PTX with that of pure PTX estimated the PEC values that indicate the amount of a drug incorporation for single cell, that was 0.094 ± 0.05 pg/cell (mean ± SD) for MSCs-PTX/LYS-FT and 0.124 ± 0.059 pg/cell (mean ± SD) for MSCs-PTX/LYS-F; the difference was not statistically significant (*p* > 0.1). Based on the amount of drug incorporated and carried by each cell, we estimated that, in order to produce 50% inhibition of CFPAC-1 proliferation, about 1.2 × 10^4^ of PTX loaded cells were required. The amount of PTX incorporated by MSCs evaluated by the biological activity on CFPAC-1 was also compared with those evaluated by HPLC analytical method. As shown in Figure 5C lower activity of the frozen samples correlated with a lower dosage by HPLC. However, these differences were not statistically significant (*p* > 0.3).

We then evaluated the in vitro antitumor potential of MSCs-PTX derived CM (MSCs-PTX/CM). Two human cell lines (human pancreatic adenocarcinoma cell line CFPAC-1 and mesothelioma cell line NCI H2052) were used. As shown in Figure 6 all MSCs-PTX/CM were able to inhibit cell proliferation of both CFPAC-1 (A) and NCI-H2052 (B). The antiproliferative capacity of MSCs-PTX/CM was equivalent to those obtained by adding pure PTX ranging from 10.99 ± 3.82 ng/mL to 17.36 ± 4.22 ng/mL for CFPAC-1 and from 15.14 ± 3.15 ng/mL to 27.35 ± 3.98 ng/mL for NCI-H2052. A high antitumor activity of MSC-PTX/CM was maintained at 48 and 72 h, decreasing only at 96 h incubation.

## 4. Discussion

As MSCs seem a very promising tool to use for cell therapy [1], the development of a standard procedure is fundamental to optimize their preparation, to reduce variability among batches, and to attain reproducible ATMPs. Therefore, the development of the most reliable and reproducible method for MSCs isolation and expansion is crucial to get adequate quantity of high-quality MSCs for cell therapy.

Previous studies, from our group and others, demonstrated that MSCs, when exposed to a significant amount of anticancer drugs, are able to uptake and subsequently release the drugs, thus becoming an effective tool to transport and deliver molecules at the tumor site [3,5,6].

Among the different drugs used to load MSCs, PTX is a very promising molecule, combining both antiproliferative and antiangiogenic activity [8,9]. We demonstrated that PTX after loading is released by MSCs both as free molecules and associated to extracellular microvesicles [10,34]. Indeed, in order to better investigate the possible morphological alterations induced on MSCs by PTX treatment and the involvement of MV in PTX delivery, a fine morphological investigation was performed, showing that morphology and sub-cellular organization of both MSCs and MSCs-PTX were similar, but an increased number of vacuole-like structures can be detected in MSCs-PTX, some of which were attributable to maturing multi-vesicular bodies; variably sized vesicles budding from or lying near the cell surface were observed. The release was achieved by fusion of MV membrane with cell membrane and exosome delivery in the extracellular environment. [3,10]. It is correct to consider that in vivo also the cell death can contribute to deliver drug in situ. Furthermore, it is also possible that the MSCs secretoma contribute to increase the anticancer action of PTX.

Regarding the in vitro anticancer activity, we found that the capacity to inhibit cancer cells proliferation either using cell lysate or the conditioned medium (CM) was not different from those previously reported by our group [3,31]. We know that there is a big difference between cell lysates or CM and viable cells, indeed we also performed previous studies in vitro by co-culturing MSCs-PTX (from different sources) and several tumor models [3,22,31], but in this study we preferred to optimize a potency test that could also be used as quality control test for batch release.

The use of MSCs to transport and delivery anticancer drugs should have several potential advantages over the free-form drugs: (a) this approach may increase the protection of the drug from degradation before reaching the target cancer cells; (b) MSCs being capable of integrating into the tumor stroma, PTX loaded MSC could be important for loco-regional treatment of solid tumors, where the delivered drug can reach effective anticancer level into the tumor environment thus increasing tumor drug uptake; (c) drug delivered by the cells can be better localized in the tumor mass by reducing its interaction with normal cells and contributing to decreasing systemic toxicity, due to the high dosage needed to inject drugs intravenously, to reach high levels into the tumor mass. On the other side, free PTX injected in situ is rapidly absorbed in blood, with a low local efficacy. The above points could explain the effectiveness of the treatment, as demonstrated in some preclinical studies carried out by our group [7].

It is true that we do not really know what the behavior of MSCs-PTX would be once transplanted into a host, when transplanted and exposed to the patient’s own biological milieu. Normal unloaded MSCs can undergo changes that alter their targeting capabilities. Unfortunately, to date, no studies in humans exist that can give us indications about this issue, nevertheless, our preliminary studies “in vivo” reassured us, as it does not appear from the data obtained a change in the behavior of MSCs-PTX transplanted in mice or rats. However, we know that these cells lose their ability to proliferate and differentiate and in our experiments demonstrated the anticancer-activity of MSCs-PTX both in vitro and in vivo in many tumor models (i.e., glioblastoma, pancreatic adenocarcinoma, pulmonary tumors) [3,22,31].

One of the major limitations to translating these results in the clinical practice remains the difficulty of producing a large amount of MSCs-PTX in a reasonably short period of time. Keeping in mind that GMP rules has recently recommended the development of closed processes to improve aseptic conditions, we speculated to use a closed bioreactor system, designed for automated cell expansion, to efficiently generate large amounts of MSCs loaded with PTX, leading to the possibility of developing a new ATMP to be used for the therapy of human cancers.

The use of bioreactors to produce clinical-grade MSCs has been already described [35,36], but the idea to use a bioreactor to develop a new ATMP that combine cells and an anticancer drug, such as PTX, is, to our knowledge, completely new.

We here describe a successful procedure that leads to producing up to several millions of MSCs-PTX in very few days. Our procedure starts from an initial injection of MSCs into bioreactor of approximately 20 million, according to previously described reports [35,36]. The initial step of MSCs isolation and expansion is performed under standard conditions, and this step, according to GMP suggestions, could be considered a technical limit of the bioreactor. However, we used this isolation/first expansion phase to select, through the choice among different MSCs preparations, those with the lowest PDT value. This choice has the advantage that only MSCs preparations with a high proliferative rate, thus those with a lower number of in vitro culture passages, were used for large-scale expansion in the bioreactor. To note, a reduction of the number of passages may reduce the risk of potential MSCs modifications, that can occur during a prolonged culture [37,38].

Regarding this, Von Bahr and colleagues recently published clinical results suggesting that acute GvHD patients treated with early passage MSCs had a better survival than those treated with late passage cells [39]. Indeed, it is now generally accepted that MSCs will begin to senesce after a certain number of cell divisions and that this is better evaluated by their PDT rather than by the number of passages or the duration of culture.

We here performed 13 different preparations of MSCs-PTX; we were able to produce at the end of the production process a mean of 591.43 × 10^6^ MSCs-PTX (range: 290 × 10^6^–940 × 10^6^ cells) in only few passages, demonstrating that loading MSCs with PTX does not cause a decrease in yield in terms of large scale cell expansion; indeed other groups, addressed to optimize large scale expansion of MSCs without loading them with drugs, described that using bioreactors it is possible to obtain a mean of 451 million cells [15,35,36].

The cells obtained through the bioreactor procedure, maintained high levels of the typical MSCs phenotypic markers (CD105, CD90, CD73) with a very low content of cells with necrotic or apoptotic features. HPLC analysis confirmed the presence of PTX in the MSCs preparations and, most importantly, either their lysate or CM were able to strongly inhibit in vitro human cancer cell proliferation. Each lot resulted very similar in terms of phenotypic features and antitumor activity (meaning also PTX incorporation), indicating a high level of reproducibility of our preparation method. In our opinion this result is very important, particularly in light of the effort of many investigators working to reduce the high variability among ATMP preparations [40].

Finally, it was essential to determine if MSCs-PTX prepared in a bioreactor could be banked in view of their allogeneic use for patient treatment scheduled over time. We evaluated the possible banking of MSCs-PTX by investigating if their function was maintained after cryopreservation; results displayed that the function and the potency, in terms of viability, recovery, and PTX release capacity of thawed cells did not differ statistically from those of fresh cells.

## 5. Conclusions

In conclusion the results of our study validate a protocol that is suitable for large scale production of MSCs loaded with PTX to be used in clinical trials for the treatment of patients with oncological diseases. Further studies will be addressed to evaluate the time freezing stability of MSCs-PTX (at least up to 12 months) and to validate the procedure under GMP condition to obtain the authorization by regulatory agency to produce a cell therapy product to enter a phase I clinical trial in patients with malignant pleural mesothelioma.

## Figures and Tables

**Figure 1 pharmaceutics-12-00411-f001:**
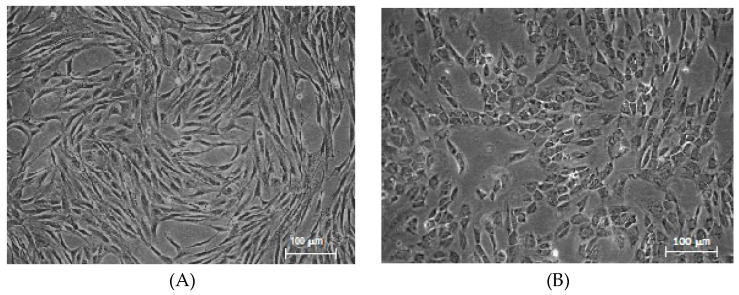
(**A**) Mesenchymal stromal cells (MSC) morphology. Spindle-shaped morphology of MSCs (P3) after the first expansion phase. Magnification 5×. (**B**) Paclitaxel-loaded Mesenchymal Stromal Cells (MSCs-PTX) morphology. Round-shaped morphology of MSCs-PTX after expansion and loading in Quantum cell expansion system. Magnification 5×.

**Figure 2 pharmaceutics-12-00411-f002:**
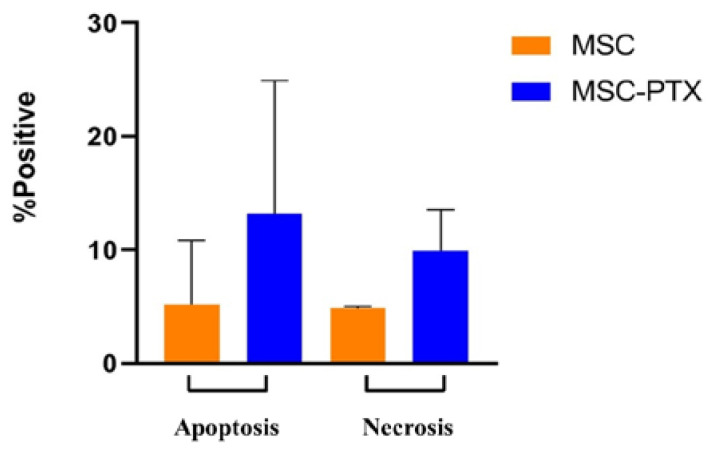
Apoptosis and necrosis. Analysis of apoptotic and necrotic cells performed by flow cytometry on MSC and MSC-PTX. The percentage of positive cells is reported as the mean ± SD of *n* = 6 experiments. No statistically significant differences were found (*p* > 0.2).

**Figure 3 pharmaceutics-12-00411-f003:**
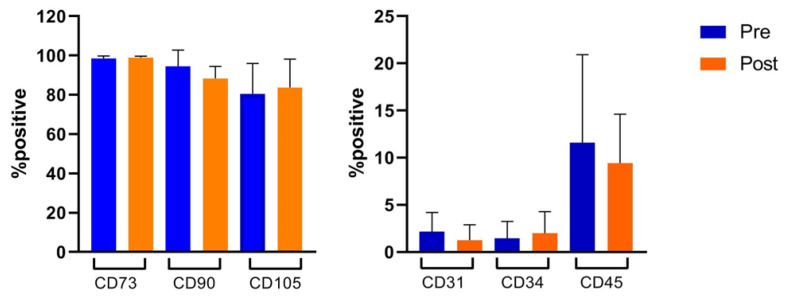
Flow cytometry analysis. Immunophenotypic characterization was performed by flow cytometry on MSC (*n* = 13) and MSC-PTX (*n* = 12) after the first expansion phase (pre) and after expansion and loading in Quantum cell expansion system (post), respectively. The percentage of positive cells is reported as the mean ± SD. No statistically significant differences were found (*p* > 0.5).

**Figure 4 pharmaceutics-12-00411-f004:**
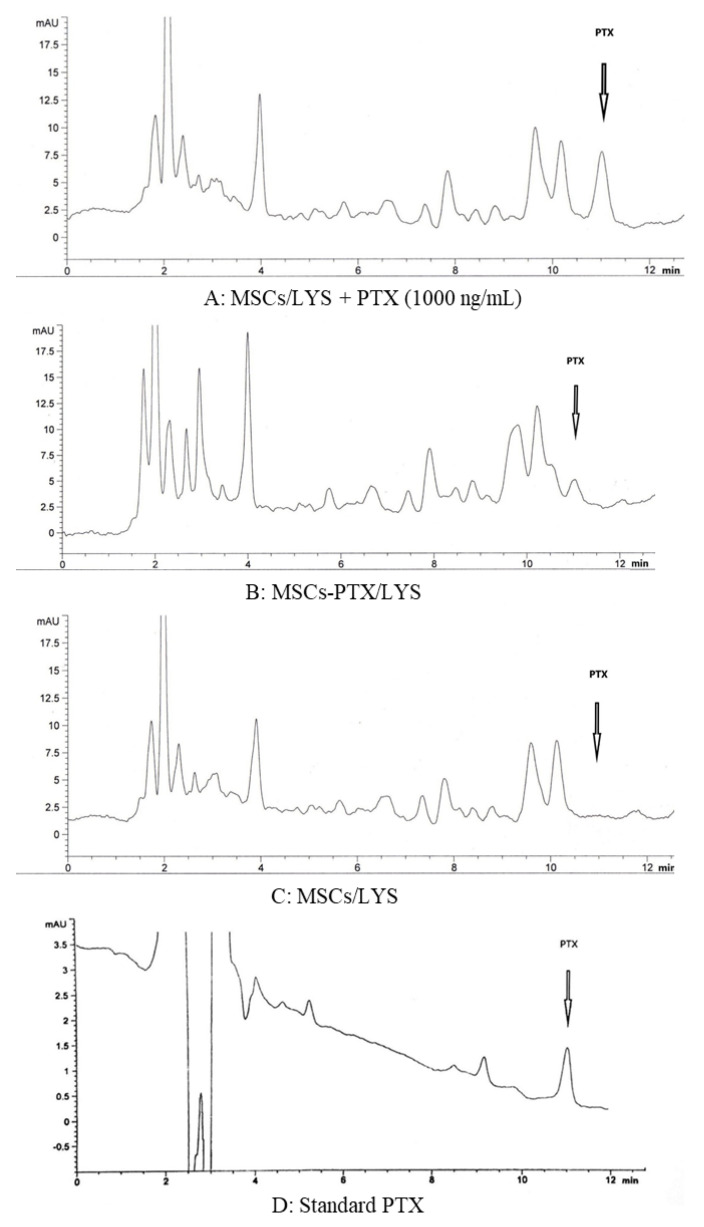
High performance liquid chromatography (HPLC) analysis of the PTX incorporated by MSCs. The figure reports the chromatogram profiles of one typical determination by HPLC analysis of the PTX incorporated by MSCs. (**A**) HPLC chromatogram of lysates from MSCs (MSCs Lysates, MSCs/LYS) plus standard drug (PTX = 1.000 ng/mL). The drug was eluted with a peak at 11 min. (**B**) HPLC chromatograms of MSCs/LYS loaded with PTX (MSCs-PTX/LYS) showing a peak of identical retention time. HPLC analysis revealed the presence of other nonspecific peaks (at 2.5–4 min) also present in the chromatogram. (**C**) HPLC chromatogram of lysate from untreated MSCs/LYS (control). (**D**) PTX standard chromatogram.

**Figure 5 pharmaceutics-12-00411-f005:**
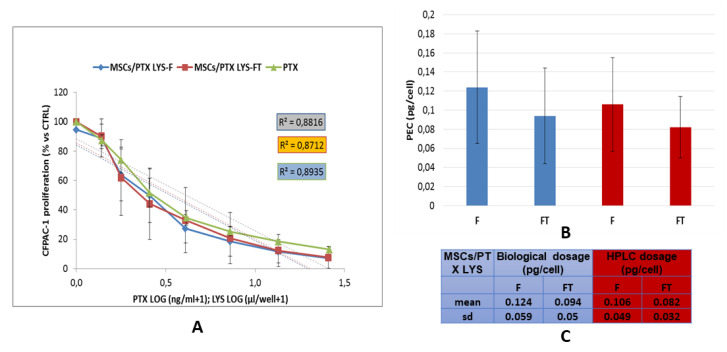
Anticancer activity of lysates of MSCs-PTX. (**A**) The graph shows the anticancer effect of MSCs-PTX/LYS expressed as percentage of CFPAC-1 proliferation referred to that of untreated tumor cells (100% proliferation). The linear regression and the correlation coefficient (R^2^) of the dose response kinetics is reported. (**B**,**C**) The histogram and the table report the PTX Equivalent Concentration (PEC) values of the drug incorporated by each MSCs (pg/mL) estimated by the biological dosage assays and by analytical assay HPLC. All the data are expressed as mean ± standard deviation (SD) of nine experiments. MSCs-PTX/LYS–F: lysates from fresh MSCs-PTX; MSCs-PTX/LYS–FT: lysates from frozen-thawed MSCs-PTX.

**Figure 6 pharmaceutics-12-00411-f006:**
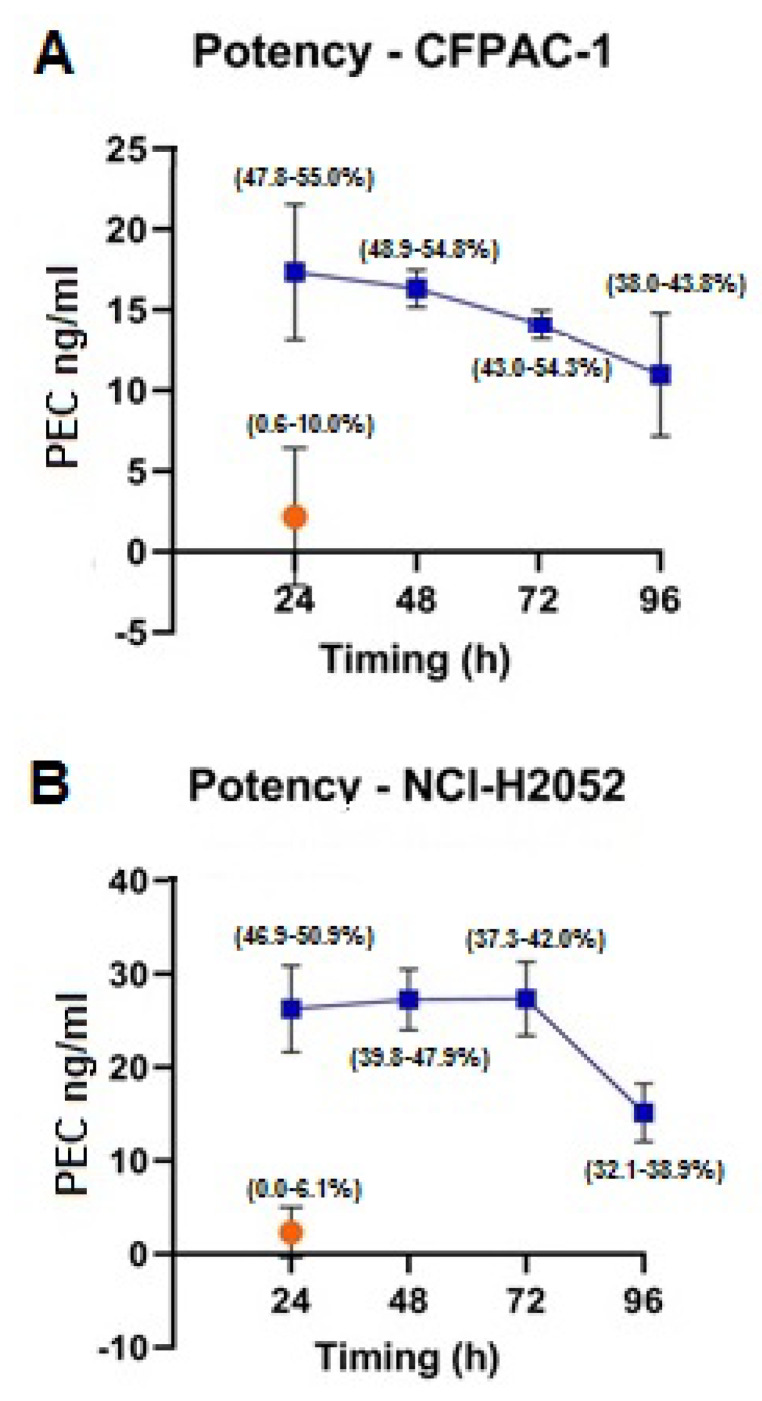
Potency. MSCs-PTX-derived Conditioned Medium (MSCs-PTX/CM) antiproliferative activity (blue line) on (**A**) the target pancreatic adenocarcinoma cells CFPAC-1 and (**B**) the mesothelioma cell line NCI H2052. Data are expressed as Paclitaxel Equivalent Concentration (PEC ng/mL, *y* axis). The *x* axis represents the time of CM collection after starting culture; the percentages of inhibition of cell line proliferation (range min–max) are delimited by parentheses (round brackets) at each time point. As shown, the antitumor activity of MSC-PTX/CM is already evident at 24 h, remained constant at 48 and 72 h, and decreased only at 96 h from initial culture. CM: orange dot represents the effect of CM from untreated MSCs (CTRL: control).

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
