# Peer review of "Automated Large-Scale Production of Paclitaxel Loaded Mesenchymal Stromal Cells for Cell Therapy Applications"

_pharmaceutics, 2020, doi:10.3390/pharmaceutics12050411_

Round 1

Reviewer 1 Report

In this manuscript, the authors successfully prepared a large amount of GMP-compliant paclitaxel (PTX)-loaded MSCs using a closed bioreactor system and evaluated the characteristics and anticancer activity of the cells. Although the cell preparation system may be attractive for MSCs-based cancer therapy, the data and descriptions are insufficient in some points. Thus, the authors should improve the quality of manuscript.

Major points

  1. The authors evaluated the viability and MSC markers of MSCs after PTX loading. However, the most important function of MSCs in cancer therapy is the tumor-homing ability as the authors mentioned in Abstract. The authors should evaluate the in vitro migration ability and in vivo tumor-homing ability of MSCs after PTX loading.

  1. While PTX loading hardly affected the viability of MSCs, the loading amount of PTX was too low. In fact, PTX-loaded MSCs took seven days to inhibit the growth of the tumor cells. Therefore, the authors should discuss the loading amount of PTX and the anticancer effect compared with the other reports.

  1. Figure 1: The appearance of MSCs were affected by PTX loading, indicating the changes in characteristics of MSCs. The authors are required to evaluate the viability for a longer period. In addition, the differentiation ability of MSCs also needs to be assessed.

  1. Figure 6 and 7: To evaluate the anticancer activity of PTX-loaded MSCs, the authors used the lysate or CM of PTX-loaded MSCs. Since there is a big difference between cell lysates or CM and viable cells, the authors should evaluate the anticancer activity by co-culturing tumor cells with PTX-loaded MSCs.

Minor points

  1. Materials and Methods: The information on reagents including antibodies is lacking. The authors should indicate the product name and company of reagents used in this manuscript.

  1. Materials and Methods: The authors described ‘multi-differentiation ability towards mesodermal lineages’, but the authors do not evaluate the differentiation of MSCs. Please delete the sentence or carry out the experiment.

  1. Figure 2: The experimental method and sample number were not shown. Please add the information.

  1. Figure 4: The authors should add the description about each line to the legend.

  1. Figure 5: To confirm the peak of PTX in HPLC analysis, the authors should add the data of PTX without cell lysate.

  1. Figure 6A: It is difficult to understand the figure. Why did the authors describe two horizontal axes? Please make it easier to understand.

  1. Figure 7: The authors showed the growth inhibition in Figure 7. Why the vertical axis is the concentration of PTX? Please make it clearer.

  1. References: The style of references is not unified. Please revise them.

Author Response

Answers to Reviewer 1

In this manuscript, the authors successfully prepared a large amount of GMP-compliant paclitaxel (PTX)-loaded MSCs using a closed bioreactor system and evaluated the characteristics and anticancer activity of the cells. Although the cell preparation system may be attractive for MSCs-based cancer therapy, the data and descriptions are insufficient in some points. Thus, the authors should improve the quality of manuscript.

Major points

1. The authors evaluated the viability and MSC markers of MSCs after PTX loading. However, the most important function of MSCs in cancer therapy is the tumor-homing ability as the authors mentioned in Abstract. The authors should evaluate the in vitro migration ability and in vivo tumor-homing ability of MSCs after PTX loading.

The comments of the reviewer are relevant. The capacity of MSCs to home cancer cells is an important functional characteristic that should be investigated. At this regard we have already demonstrated that MSCs-PTX still retain the capacity to home cancer site (Pessina A et al. Br J Haematol. 2013 Mar 160(6):766-78; Pessina A et al. J Exp Clin Cancer Res. 2015 Aug 13 34:82; Pacioni S. et al. Stem Cell Res Ther. 2017 Mar 9;8(1):53). Furthermore, in these previous studies we have seen that only 10% of the systemically inoculated cells reach the tumor site, suggesting that, to improve therapeutic efficacy, higher dose of cells are probably required. Therefore, based on these previous studies we thought that in order to move towards MSCs-PTX clinical application, it was necessary to develop a rapid method for cells large-scale production under GMP rules.

Regarding migration of MSCs-PTX, we have found that once loaded with PTX, MSCs reduced their migration but maintained the capacity to attract and kill cancer cells (Pessina A et al Br J Haematol. 2013 Mar;160(6):766-78). We have evaluated the capacity of MSCs-PTX to block migration cancer cells (data not included in the paper): both the addition of MSCs-PTX or their CM were able to block cancer cells (MSTO mesothelioma cell line) migration in a 3D collagen gel assay, suggesting that localization of MSC-PTX nearby the tumoral nodules, besides proliferation, may also block cancer cells migration (reducing tumor infiltration).

2. While PTX loading hardly affected the viability of MSCs, the loading amount of PTX was too low. In fact, PTX-loaded MSCs took seven days to inhibit the growth of the tumor cells. Therefore, the authors should discuss the loading amount of PTX and the anticancer effect compared with the other reports.

Probably the reviewer made a mistake because our results show that viability of MSCs was not affected by the treatment with PTX (see the Results section, paragraph: “MSC expansion and loading with PTX in Quantum system”). However, as expected, we found that MSCs after PTX loading changed their  morphology (transition from the typical spindle-shaped morphology to round-shaped cells, see Fig.1 and Supplementary Fig.S2) and lost proliferative capacity. The amounts of PTX incorporated by MSCs, using our bioreactors procedure, were 0.094±0.05 pg/cell (mean±SD) for MSCs-PTX/LYS after freeze/thawing and 0.124±0.059 pg/cell (mean±SD) for fresh MSCs-PTX, as described in the Results section paragraph “In vitro anticancer activity of MSCs-PTX ” and it was on line with what we have previously obtained following a standard culture and loading procedure (Pessina A et al. PLoS One. 2011;6(12):e28321; Bonomi A. Et al Int J Immunopathol Pharmacol. 2013 Jan-Mar;26(1 Suppl):33-41). In addition, we found that the capacity to inhibit cancer cells proliferation either using cell lysate or the conditioned medium (CM) (see Fig.5 and Fig.6 of the manuscript) was not different from those previously reported by our group (Pessina A et al. PLoS One. 2011;6(12):e28321; Bonomi A. Et al Int J Immunopathol Pharmacol. 2013 Jan-Mar;26(1 Suppl):33-41).  The anti-proliferative activity of MSCs-PTX/CM was already evident at 24h and was maintained even after 72h (see Fig.6), suggesting a slow release of the drug by the cells, persistent at least for three days.

3. Figure 1: The appearance of MSCs were affected by PTX loading, indicating the changes in characteristics of MSCs. The authors are required to evaluate the viability for a longer period. In addition, the differentiation ability of MSCs also needs to be assessed.

Of course this is an important point that we discussed also in previous reports starting from the procedure of MSCs expansion and loading in flasks (Pessina A et al. PLoS One. 2011;6(12):e28321).The viability of MSCs after expansion/loading with PTX in Quantum Cell Expansion System was evaluated at different time points; in details MSCs-PTX, both fresh and after freeze/thawing, were re-seeded in T25 flasks (3 flasks/condition, total six flasks/experiment) and the MSCs-PTX viability was analyzed after 7, 14 and 21 days. In this period of time medium was changed every 3 days; the cells have never been detached, due to the lost of their duplication capacity and the failure to reach confluence. Results of n=3 experiments are shown in Supplementary Figure S1. MSCs-PTX maintain their viability for at least 21 days, however they lost the capacity to proliferate and differentiate. This in term of a biological function represents a limitation. On the contrary this is a significant advantage by considering the role that MSCs-PTX must play, that is to act as a carrier able to deliver the drug where are injected or are able to home. This is of great importance, because contributes to reduce the risks and supports the safety of the MSCs-PTX, that can not escape to give abnormal proliferation or ectopic differentiation when injected into the host.

A paragraph regarding this issue has been added in the “Materials and Methods” (see pages 4, lines 145-149) and in “Results” (see page 7, lines 261-262).

4. Figure 6 and 7: To evaluate the anticancer activity of PTX-loaded MSCs, the authors used the lysate or CM of PTX-loaded MSCs. Since there is a big difference between cell lysates or CM and viable cells, the authors should evaluate the anticancer activity by co-culturing tumor cells with PTX-loaded MSCs.

To confirm the direct role of PTX loaded MSCs we performed many studies in vitro by co-culturing MSCs-PTX (from different sources) and several tumor models. (Pessina A et al. PLoS One. 2011;6(12):e28321; Bonomi A et al. Int J Immunopathol Pharmacol. 2013 Jan-Mar;26(1 Suppl):33-41; Brini AT et al. Expert Opin Drug Deliv. 2016, 13(6), 789-98). However we stated this fact in a sentence in the “Discussion” section, page 13, lines 392-397.

Minor points

1. Materials and Methods: The information on reagents including antibodies is lacking. The authors should indicate the product name and company of reagents used in this manuscript.

The information on reagents were implemented in the text of the manuscript.

2. Materials and Methods: The authors described ‘multi-differentiation ability towards mesodermal lineages’, but the authors do not evaluate the differentiation of MSCs. Please delete the sentence or carry out the experiment.

The differentiation ability of MSCs was evaluated on the MSCs (non-loaded cells) to confirm that the production under experimental conditions does not modify the phenotypic/functional pattern of the cells. This represents a criterion of quality in MSCs expansion procedure. The PTX loaded MSCs do not proliferate and are unable to differentiate (see answer to point 3). However the data was not presented in this manuscript, addressed mainly to characterize MSCs-PTX, therefore the sentence was deleted. 

3. Figure 2: The experimental method and sample number were not shown. Please add the information.

“Material and Methods” section was implemented with the description of method used to assess apoptosis and necrosis (Material and Methods section, page 4, lines 151-161); sample number was added in the legend of Figure 2. Moreover results of the apoptotic and necrotic cell populations detected by Annexin V-FITC and PI staining of a representative sample of MSCs and the correspondent MSCs-PTX were added as Supplementary Figure S3.

4. Figure 4: The authors should add the description about each line to the legend.

Figure 4 was moved to Supplemental data (Supplementary Figure S4) and the description of each line was added to the Legend.

5. Figure 5: To confirm the peak of PTX in HPLC analysis, the authors should add the data of PTX without cell lysate.

We did not present the HPLC standard profile in solvent but only the profile of PTX added to lysate of untreated MSCs, in order to demonstrate that in the lysate the drug had the same elution time. However the reviewer comment is right. So we insert in the Figure 4 (ex Figure 5) the standard elution profile of PTX.

6. Figure 6A: It is difficult to understand the figure. Why did the authors describe two horizontal axes? Please make it easier to understand.

One of the two horizontal axes in figure 5A (ex Figure 6A) has been deleted.

7. Figure 7: The authors showed the growth inhibition in Figure 7. Why the vertical axis is the concentration of PTX? Please make it clearer.

The Figure 6 (ex Figure 7) and the correspondent Legend have been clarified.    

8. References: The style of references is not unified. Please revise them.

The style of “References” section has been revised.

Reviewer 2 Report

Pharmaceutics_780094

Automated large-scale production of paclitaxel loaded MSCs for cell therapy applications

In this work, the authors produce high amounts of MSCs in a close bioreactor system under GMPs conditions. The MSCs are loaded with the chemotherapeutic drug paclitaxel (PTX) in order to be used as a cell therapy approach to cancer patients. Importantly, the authors have isolated MSCs from 13 different donors, a work that must be recognized. PTX-MSCs lysates and conditioned medium from these cells are shown to inhibit the proliferation tumor cell lines in vitro. The culture of the MSCs in the bioreactor as well as their treatment with PTX seems to be efficient. However, there are some missing points (technical, conceptual) that should be addressed in order to improve the quality of the manuscript.

Introduction:

1.-Please, provide more feedback regarding the use of MSCs as chemotherapeutic vehicles, in my opinion this subject, which is the main one in this manuscript (apart from obtaining a high number of MSCs under GMPs) is not explained enough.

Lane 45: Please, give an explanation of paclitaxel functions in the cell: targets, consequences for the cell …

Line 49: Same comment for Sorafenib.

Results:

2.-Figure 1: Authors show images of MSCs isolated from lipoaspirates (1A) and MSCs after expanded in the bioreactor and treated with PTX (1B). The change in MSCs morphology is remarkable. Since authors can not follow the cell morphology during the amplification period, how do they know that it is not due to the process of expansion per se? It would be more informative regarding this change in morphology if authors show the following images:  1) MSCs cultured in plastic (treated previously with fibronectin) with and without PTX and 2) MSCs expanded in the bioreactor with and without PTX.

Please, include scale bars in the images.

3.- Line 231-232: “….. and los the growth capacity (Fig 1B). Figure 1 B just reflects the change in cell morphology, authors can not state for a lost in growth capacity from this image. Please, re-write this sentence.

4.- Figure 2. The authors claim that they used flow cytometry experiments to analyze apoptosis/necrosis in MSCs after being expanded in the bioreactor (with or without PTX). However, there is no description about the method used (Anexin V/IP?). Please, include a protocol in Mat and Met.  The graphs (dotplots?) obtained from those flow cytometry experiments should be included as Suplementary information.

5.- Figure 3. The percentage of CD45 positive cells is quite high, they should be <2% positive as established by ISSCR to be defined as MSCs. Please, give an explanation for this observation.

6.-Figure 4. The information given in this Figure is not necessary. It is already reflected in Figure 3. Please, move it to Supplemental data.

7.-The author demonstrate the loading of PTX into MSCs by HPLC experiments. The lysates from these cells are shown to have anti-cancer properties in tumor cell lines. This is supposed to be mediated by the presence of PTX in the lysates. Then, authors show that the CM from MSCs-PTX inhibits also the proliferation of tumor cell lines.

-which would be the mechanism in this case? Is it presumed that PTX is secreted by cells to the CM? If so, authors must detect the presence of PTX in the conditioned medium by HPLC. In fact, if the final purpose of this work is to offer a cell therapy product with anti-tumoral properties, it is not probably that the administerd MSCs are going to be lysed in the host to inhibit then the growth of tumor cells. Due to the main paracrine role of MSCs, it would be very likely that they would exert this hypothetical anti-tumor function due to paracrine secretion. It would be of interest then, to try to detect PTX in the CM of MSCs-PTX. It could be posible also, that as a consequence of treating MSCs with PTX, they change the composition of their normal secretome. In consequence, an analysis of the composition of this conditioned medium would be very valuable.

Please, include a paragraph in the discussion regarding this issue.

8.-Miscellaneous questions that should be discussed:

-Which would be the advantage of treating patients with paclitaxel-MSCs instead of paclitaxel itself? I think that this is a very important poit that should be developed and discussed.

-There is controversy in the literature regarding the treatment of MSCs with chemotherapic drugs: there are works describing just the opposite results: chemotherapy-educated MSCs leads to the enrichment of the tumor iniciating cells in specific tumor types. When injectated in tumor-bearing mice, cells increased the rate of tumor growth. (Timaner M et al., 2018).

-Once transplanted and exposed to the patient's own biological milieu, MSCs undergo changes that alter their targeting capabilities. Authors do not really know which would be the behaviour of MSCs-PTX once transplanted into a host.

Author Response

Answers to Reviewer 2

In this work, the authors produce high amounts of MSCs in a close bioreactor system under GMPs conditions. The MSCs are loaded with the chemotherapeutic drug paclitaxel (PTX) in order to be used as a cell therapy approach to cancer patients. Importantly, the authors have isolated MSCs from 13 different donors, a work that must be recognized. PTX-MSCs lysates and conditioned medium from these cells are shown to inhibit the proliferation tumor cell lines in vitro. The culture of the MSCs in the bioreactor as well as their treatment with PTX seems to be efficient. However, there are some missing points (technical, conceptual) that should be addressed in order to improve the quality of the manuscript.

Introduction:

1.-Please, provide more feedback regarding the use of MSCs as chemotherapeutic vehicles, in my opinion this subject, which is the main one in this manuscript (apart from obtaining a high number of MSCs under GMPs) is not explained enough.

Lane 45: Please, give an explanation of paclitaxel functions in the cell: targets, consequences for the cell …

Line 49: Same comment for Sorafenib.

The “Introduction” section has been implemented, as request. (see page 2, lines 45-62)

Results:

2.-Figure 1: Authors show images of MSCs isolated from lipoaspirates (1A) and MSCs after expanded in the bioreactor and treated with PTX (1B). The change in MSCs morphology is remarkable. Since authors can not follow the cell morphology during the amplification period, how do they know that it is not due to the process of expansion per se? It would be more informative regarding this change in morphology if authors show the following images:  1) MSCs cultured in plastic (treated previously with fibronectin) with and without PTX and 2) MSCs expanded in the bioreactor with and without PTX.

Please, include scale bars in the images.

The images were implemented as requested (see Supplementary Figure S2). Moreover supplementary data on viability of MSCs have been added. The viability of MSCs after expansion/loading with PTX in Quantum Cell Expansion System was evaluated at different time points; in details MSCs-PTX, both fresh and after freeze/thawing, were re-seeded in T25 flasks (3 flasks/condition, total six flasks/experiment) and the MSCs-PTX viability were analyzed after 7, 14 and 21 days. In this period of time medium was changed every 3 days; the cells have never been detached, due to the lost of their duplication capacity and the failure to reach confluence. Results of n=3 experiments are shown in Supplementary Figure S1.

Scale bars were included in the images.

3.- Line 231-232: “….. and los the growth capacity (Fig 1B). Figure 1 B just reflects the change in cell morphology, authors can not state for a lost in growth capacity from this image. Please, re-write this sentence.

The sentence has been changed (see page 7, lines 268-269).

4.- Figure 2. The authors claim that they used flow cytometry experiments to analyze apoptosis/necrosis in MSCs after being expanded in the bioreactor (with or without PTX). However, there is no description about the method used (Anexin V/IP?). Please, include a protocol in Mat and Met.  The graphs (dotplots?) obtained from those flow cytometry experiments should be included as Suplementary information.

“Material and Methods” section was implemented with the description of method used to assess apoptosis and necrosis. (see page 4, lines 151-161).

Moreover results of the apoptotic and necrotic cell populations detected by Annexin V-FITC and PI staining of a representative sample of MSCs and the correspondent MSCs-PTX were added as Supplementary Figure S3.

5.- Figure 3. The percentage of CD45 positive cells is quite high, they should be <2% positive as established by ISSCR to be defined as MSCs. Please, give an explanation for this observation.

We absolutely agree with the reviewer comment that is pertinent. Our results show a percentage of the CD45 marker higher than that established by ISSCR to define cells as MSCs. On this point previous papers displayed conflicting evidences; in most cases the percentage of CD45 was below 2%, but some authors have found different results. Okolicsanyi et al. (PLoS One. 2015 Sep 10;10(9):e0137255) found a high percentage of the CD45 haematopoietic marker not only at early but also at late MSCs culture passages. Some evidence suggests the co-existence during MSCs culture of haematopoietic stem cells with MSCs over time, that in part can be explained by the presence of different CD45 isoforms as shown in the manuscript of Penninger JM et al. (Nat Immunol. 2001 May;2(5):389-96).

However, there remains inconsistency regarding the accepted cell surface profile of MSCs and this point needs further studies in the future.

6.-Figure 4. The information given in this Figure is not necessary. It is already reflected in Figure 3. Please, move it to Supplemental data.

Figure 4 has been moved in the Supplementary material (Supplementary Figure S4).

7.-The author demonstrate the loading of PTX into MSCs by HPLC experiments. The lysates from these cells are shown to have anti-cancer properties in tumor cell lines. This is supposed to be mediated by the presence of PTX in the lysates. Then, authors show that the CM from MSCs-PTX inhibits also the proliferation of tumor cell lines.

-which would be the mechanism in this case? Is it presumed that PTX is secreted by cells to the CM? If so, authors must detect the presence of PTX in the conditioned medium by HPLC. In fact, if the final purpose of this work is to offer a cell therapy product with anti-tumoral properties, it is not probably that the administerd MSCs are going to be lysed in the host to inhibit then the growth of tumor cells. Due to the main paracrine role of MSCs, it would be very likely that they would exert this hypothetical anti-tumor function due to paracrine secretion. It would be of interest then, to try to detect PTX in the CM of MSCs-PTX. It could be posible also, that as a consequence of treating MSCs with PTX, they change the composition of their normal secretome. In consequence, an analysis of the composition of this conditioned medium would be very valuable.

Please, include a paragraph in the discussion regarding this issue.

The kinetics of drug release in the conditioned medium had been previously described by our group (Pessina et al PLoS One. 2011, 6(12), e28321). We demonstrated that MSCs can acquire strong anti-tumor activity after priming with PTX through their capacity to uptake and then release the drug in the CM; as supposed by the reviewer we found that in vitro PTX is released by MSCs-PTX both as free molecule and associated to extracellular microvesicles (Pascucci et al. Journal Controlled Release 2014 192: 262-270; Coccè et al . Pharmaceutics 2019, 1; 11 (2)). The presence of PTX in both CM and MV was confirmed by HPLC analysis; moreover we assessed the presence of PTX in MV by using FTIR spectroscopy, a method that does not alter MV structures, and the results strongly suggested that analyzed MV were loaded with PTX. In order to better investigate the possible morphological alterations induced on MSCs by PTX treatment and the involvement of MV in PTX delivery, a fine morphological investigation was also performed by transmission electron microscopy (TEM): the analysis showed that morphology and sub-cellular organization of both MSCs and MSCs-PTX were similar, but an increased number of vacuole-like structures was detected in MSCs-PTX, some of which were attributable to maturing multi-vesicular bodies; variably sized vescicles budding from or lying near the cell surface were observed. The release was achieved by fusion of MV membrane with cell membrane and exosome delivery in the extracellular environment. This concern was confirmed also analyzing MSCs-PTX over time by confocal microscopy using fluorescent PTX (PTX-F): after a hour of priming the internalization of PTX-F by MSCs was appreciable, the staining was intense and enriched inside MV; after others 24h we observed that  the distribution of PTX-remained confined to vescicles, many of which were closed to the cell membrane, suggesting the secretion of PTX from MV.  (Pascucci et al. J Contr Release. 2014, 192, 262-270; Pessina et al. PLoS One. 2011, 6(12), e28321). It is correct to consider that in vivo also the cell death can contribute to deliver drug in situ. Furthermore, it is also possible that the MSCs secretoma contribute to increase the anticancer action of PTX.

A paragraph regarding this issue has been added in the Discussion (see pages 12-13, lines 381-391).

8.-Miscellaneous questions that should be discussed:

-Which would be the advantage of treating patients with paclitaxel-MSCs instead of paclitaxel itself? I think that this is a very important point that should be developed and discussed.

A paragraph regarding this issue has been added in the Discussion (see page 13, lines 398-408)

-There is controversy in the literature regarding the treatment of MSCs with chemotherapic drugs: there are works describing just the opposite results: chemotherapy-educated MSCs leads to the enrichment of the tumor iniciating cells in specific tumor types. When injectated in tumor-bearing mice, cells increased the rate of tumor growth. (Timaner M et al., 2018).

Of course this is a hard discussed topic. We suggest the reviewer to consider some main aspects:

  1. Also the cited authors (Timaner et al ) clearly write that “…..MSC-dependent TIC enrichment occurs in pancreatic but not lung cancer cells, even though both tumor types express the CXCL10 receptor, CXCR3. ………It is plausible that the differential expression of CXCR3 isoforms in different tumor types explains the variable effect on enrichment of TICs in such tumors, as we also demonstrated in our study when using different pancreatic cancer cell lines”. This means that different drugs (eg. PTX used in our study) cannot induce the TIC enrichment described.
  2. Loco-regional treatment attain a “in situ” high drug concentration, able to work for days by favouring the PTX penetration into cancer cells. In fact, as reported by Khu et al, (1999) the high tumour cell density may represent a barrier to paclitaxel but, although the drug penetration is confined to the periphery, the apoptosis produced a reduction of epithelial cell density by favouring the drug penetration in solid tumour. A slow increasing concentration of PTX delivered by our treatment can favour the mechanism of paclitaxel penetration into the inner parts of the residual tumour masses where, according to Weaver et al ,2014 also lower concentrations, which are slower to evoke cell death, are clinically relevant.
  3. The so called education of MSCs able to produce TIC enrichment must be better defined in order to compare and discuss data. In fact, our procedure used PTX (that was not used by Timaner et al.), furthermore we used the drug at very high dosage to prime MSCs. This produce a block of main functions, included proliferation/differentiation and the primed cells survive only few time (not more than 45 days).
  4. As reported by Timaner et al their study is based on some data of cancer induced resistance described by Roodhart et al. Cancer Cell. 2011; 20(3):370–83. Without entering in detail, in our opinion these data suffer of some methodological bias that needing to be discussed. The most important regards the study of the resistance of cancer cells to platinum. It is known that Pt is very instable and also in vitro loss its activity in few hours (Rimoldi et al. Uptake-release by MSCs of a cationic platinum(II) complex active on in vitro human malignant cancer cell lines Biomedicine & Pharmacotherapy 108 (2018) 111–118). In our opinion this makes very weak some results on the increased resistance of cancer cells, that could be only the results of a decreased anticancer activity of Pt.

-Once transplanted and exposed to the patient's own biological milieu, MSCs undergo changes that alter their targeting capabilities. Authors do not really know which would be the behaviour of MSCs-PTX once transplanted into a host.

A paragraph regarding this issue has been added in the Discussion (see page 13 lines 409-417).

Round 2

Reviewer 1 Report

The authors have revised the manuscript as the response to the reviewer’s comments. The article seems more reasonable than the previous one.

Reviewer 2 Report

The authors have successfully addressed the suggested points.